# Different HSP90 Inhibitors Exert Divergent Effect on Myxoid Liposarcoma In Vitro and In Vivo

**DOI:** 10.3390/biomedicines10030624

**Published:** 2022-03-07

**Authors:** Christoffer Vannas, Lisa Andersson, Soheila Dolatabadi, Parmida Ranji, Malin Lindén, Emma Jonasson, Anders Ståhlberg, Henrik Fagman, Pierre Åman

**Affiliations:** 1Sahlgrenska Center for Cancer Research, Department of Laboratory Medicine, Institute of Biomedicine, Sahlgrenska Academy, University of Gothenburg, SE-405 30 Gothenburg, Sweden; christoffer.vannas@gu.se (C.V.); lisa.andersson.3@gu.se (L.A.); soheila.dolatabadi@gu.se (S.D.); parmida.ranji@gu.se (P.R.); malin.linden@gu.se (M.L.); emma.jonasson@gu.se (E.J.); anders.stahlberg@gu.se (A.S.); 2Department of Oncology, Sahlgrenska University Hospital, SE-413 46 Gothenburg, Sweden; 3Region Västra Götaland, Department of Clinical Genetics and Genomics, Sahlgrenska University Hospital, SE-413 45 Gothenburg, Sweden; 4Wallenberg Centre for Molecular and Translational Medicine, University of Gothenburg, SE-405 30 Gothenburg, Sweden; 5Department of Clinical Pathology, Sahlgrenska University Hospital, SE-413 45 Gothenburg, Sweden

**Keywords:** myxoid liposarcoma, HSP90 inhibition, receptor tyrosine kinase signaling, drug treatment, combination therapy

## Abstract

The therapeutic options for patients with relapsed or metastatic myxoid liposarcoma (MLS) remain scarce and there is currently no targeted therapy available. Inhibition of the HSP90 family of chaperones has been suggested as a possible therapeutic option for patients with MLS. However, the clinical effect of different HSP90 inhibitors vary considerably and no comparative study in MLS has been performed. Here, we evaluated the effects of the HSP90 inhibitors 17-DMAG, AUY922 and STA-9090 on MLS cell lines and in an MLS patient-derived xenograft (PDX) model. Albeit all drugs inhibited in vitro growth of MLS cell lines, the in vivo responses were discrepant. Whereas 17-DMAG inhibited tumor growth, AUY922 surprisingly led to increased tumor growth and a more aggressive morphological phenotype. In vitro, 17-DMAG and STA-9090 reduced the activity of the MAPK and PI3K/AKT signaling pathways, whereas AUY922 led to a compensatory upregulation of downstream ERK. Furthermore, all three tested HSP90 inhibitors displayed a synergistic combination effect with trabectidin, but not with doxorubicin. In conclusion, our results indicate that different HSP90 inhibitors, albeit having the same target, can vary significantly in downstream effects and treatment outcomes. These results should be considered before proceeding into clinical trials against MLS or other malignancies.

## 1. Introduction

Myxoid liposarcoma (MLS) is the second most common subtype of liposarcoma [1]. It is characterized by t(12;16) or less commonly t(12;22) chromosome translocations that result in the fusion oncogenes *FUS-DDIT3* or *EWSR1-DDIT3* [2,3,4]. *FUS* (also known as *TLS*) and *EWSR1* are house-keeping genes involved in RNA processing, whereas *DDIT3* encodes a stress-response transcription factor. The fusion oncoproteins FUS-DDIT3 or EWSR1-DDIT3 drive tumorigenesis in MLS, by acting as abnormal transcription factors and by interacting with chromatin modulating complexes such as the SWI/SNF complex [5,6,7].

The prognosis of MLS is generally good [8,9]. However, a subset of the patients suffer a more aggressive disease characterized by increased numbers of round cells, increased cell density and higher proliferation rates [10]. The general treatment for MLS is surgical resection, most frequently with addition of either neoadjuvant, adjuvant or in combination, radiation therapy. Metastatic MLS is treated with chemotherapy, primarily with doxorubicin [1]. Although the benefit is controversial, doxorubicin is often combined with ifosfamide to increase the activity [11,12,13,14]. Eribulin or trabectidin are used as second line treatments [15,16,17,18,19]. Trabectidin is also under investigation as a neoadjuvant treatment; a preoperative regime of trabectidin and radiation therapy has been shown to have a favorable safety profile [20] and there is currently an ongoing phase 2 clinical trial to evaluate the safety and efficacy of this treatment approach (ClinicalTrials.gov Identifier: NCT02275286).

MLS has been reported to have increased activity of receptor tyrosine kinases (RTKs), such as RET, MET, PDGFR-B, IGFR and ERBB3 [21,22,23]. Some of these RTKs crosstalk and activate each other, and ultimately trigger downstream signaling, such as the PI3K/AKT and MAPK pathways [21]. However, no therapy targeting these pathways has yet proven effective against MLS in clinical trials [21,24,25].

The heat shock protein 90 (HSP90) family of proteins has been suggested as a potential drug target in MLS [22,25]. The HSP90 family of proteins are chaperones, involved in correction and restoration of de novo synthesized or misfolded proteins, but also in formation of protein complexes. More than 700 proteins are clients of HSP90, many linked with essential cancer hallmarks, such as cell cycle progression, RTK signaling, angiogenesis and metastasis [26,27]. HSP90 normally constitutes 1–2% of the total mammalian cellular protein load and its expression is upregulated in pathological conditions such as inflammation or cancer [27]. Given the important role of HSP90 in cell survival and cell proliferation, HSP90 inhibitors have been extensively tested as cancer treatments [28]. Many HSP90 inhibitors have failed to show clinical utility due to insufficient effects or unacceptable toxicities. However, fifteen HSP90 inhibitors remain in phase 1–2 clinical trials against different forms of malignancies [28,29].

In MLS, HSP90 has been shown to potentiate proliferative signals of the RET, ERBB3 and EGFR RTKs. We previously showed that inhibition of HSP90 with 17-DMAG reduced RET and ERBB3 signaling and resulted in cell death in vitro and in vivo [22]. Another publication by Steinmann et al. reported in vitro efficacy of the HSP90 inhibitor AUY922 in the treatment of myxoid and undifferentiated liposarcomas [25]. In addition, a high-throughput drug screening on MLS cell lines identified four HSP90 inhibitors among the 27 most potent drugs against MLS [24]. Taken together, these reports suggest that HSP90 inhibition could be a potential therapeutic option in MLS.

The clinical effect and toxicity profiles between different HSP90 inhibitors display a significant variability. However, no comparative study on the effect of different HSP90 inhibitors has been performed in the treatment of a single tumor entity. Here, we have evaluated the in vitro and in vivo effects of the three HSP90 inhibitors 17-DMAG, AUY922 and STA-9090 in MLS. These HSP90 inhibitors belong to different classes; 17-DMAG is a geldanamycin derivate whereas AUY922 and STA-9090 are second generation radicicol derivates. They all target the ATP-binding pocket of HSP90 [30]. AUY922 and STA-9090 are in phase 1–2 clinical cancer trials, whereas 17-DMAG is currently not in any clinical trial [29].

## 2. Materials and Methods

### 2.1. Tumor Cell Lines and Cell Culture

MLS cell lines 402-91, 2645-94 and 1765-92 were established from MLS tumor tissue as previously described [2]. The fibrosarcoma cell line HT1080 was obtained from ATCC (CCL-121, Manassas, VA, USA). The stable transfected clone HT1080 FUS-DDIT3-EGFP was established as previously described [31] and maintained by addition of 500 µg/mL geneticin. Human fibroblasts (F470) were derived from foreskin of an anonymous donor. All tumor cell lines were cultured in RPMI1640 medium GlutaMAX with addition of 5% fetal bovine serum and 1% penicillin (100 U/mL) and streptomycin (100 µg/mL). F470 were cultured in RPMI 1640 GlutaMAX with 10% fetal bovine serum and 1% penicillin (100 U/mL) and streptomycin (100 µg/mL). All media and supplements were obtained from Gibco (Thermo Fisher Scientific, Waltham, MA, USA). Cells were incubated at 37 °C with 5% CO_2_. All cell lines were routinely screened for *Mycoplasma* infections.

### 2.2. Chemical Compounds

The HSP90 inhibitors AUY922 (luminespib), 17-DMAG (alvespimycin), STA-9090 (ganetespib), as well as doxorubicin and trabectidin (all Selleckchem, Munich, Germany) were dissolved in 100% dimethyl sulfoxide (DMSO) to a stock solution of 200 mM and stored at −80 °C. For in vitro use, drugs were diluted in phosphate-buffered saline (PBS; Thermo Fisher Scientific, Waltham, MA, USA) to a concentration of 1 mM and stored at −20 °C in a freezer. For in vivo experiments, drugs were dissolved in a 25 °C drug dilution buffer containing PBS with 5% glucose and 0.2% Tween 80.

### 2.3. Cell Viability Assays

AlamarBlue cell viability assay (Invitrogen, Carlsbad, CA, USA), according to manufacturer’s instructions, was used to determine the cytotoxic effect of HSP90 inhibitors or doxorubicin in vitro. After 72 h drug treatment, cells were analyzed spectrofluorometrically, using a Victor^3^ 1420 Multilabel Counter (PerkinElmer, Waltham, MA, USA) with an excitation frequency of 544 nM and an emission frequency of 615 nM. IC50 values were estimated from sigmoidal non-linear regression curves, using GraphPad Prism v9.1.0 (GraphPad Software Inc, San Diego, CA, USA). Six replicates were used for each drug concentration in every cell line. All experiments were run twice.

For combination drug screenings, a drug matrix with four different concentrations of each drug was used. Three technical replicates were used for each individual dose combination. Synergy was calculated using Synergy Finder web application [32] and presented as dose-response curves, synergy matrixes and a cumulative ZIP synergy score for each drug combination.

### 2.4. In Vivo Experiments

All experiments were performed in accordance with EU directive 2010/63 (Regional animal ethics committee of Gothenburg approval Dnr 129-2016, 2016-12-14). Four to six-week-old BALB/C nude mice (Janvier Labs, Le Genest-Saint-Isle, France) were transplanted with 2 × 2 mm MLS tumor pieces. The MLS tumor used in our PDX model was generously provided by Ola Myklebost—University of Bergen, Norway. When tumors had reached a size of at least 5 × 5 mm, treatment was initiated. Treatment was performed using intraperitoneal injection for 5 weeks in the following doses: 17-DMAG 25 mg/kg three days per week, AUY922 20 mg/kg three days per week and STA-9090 25 mg/kg twice per week. Measurement of tumor sizes was performed with a caliper, measuring the longest (D) and the perpendicular (d) diameter and converting into tumor volume (V) using the formula $V=\frac{\pi }{6}\times D\times d^{2}$. Tumor sizes and weights of mice were measured twice per week. Each treatment group consisted of at least 8 tumors. Tumor growth curves were generated in GraphPad Prism 9.1.0 (GraphPad, San Diego, CA, USA) after baseline correction.

Mice were sacrificed by isoflurane anesthesia, followed by incision of the right ventricle. After euthanizing, tumors were extracted and half of the tumor was put in formaldehyde for sectioning. The other half was stored in Qiagen Allprotect Tissue Reagent (Qiagen, Hilden, Germany) for long term storage and subsequent extraction of proteins or RNA. For protein extraction, frozen PDX tumors in Qiagen Allprotect tissue reagent were pulverized using a scalpel and thereafter lysed in ice cold RIPA lysis buffer supplemented with 5 mM EDTA and 1× Halt protease and phosphatase inhibitor cocktail (Thermo Fisher Scientific, Waltham, MA, USA). Tissue protein lysates were obtained by homogenization using a TissueLyzer II (Qiagen, Hilden, Germany). For RNA extraction, tumor pieces were homogenized using a TissueLyzer II (Qiagen, Hilden, Germany), followed by extraction using RNeasy Micro Kit (Qiagen, Hilden, Germany) according to manufacturer’s instructions. RNA quality was controlled with a DNF-471 RNA kit on a Fragment Analyzer (both Agilent Technologies, Santa Clara, CA, USA) according to manufacturer’s instructions. PROsize3 software (Agilent Technologies, Santa Clara, CA, USA) was used for data analysis.

### 2.5. Tumor Sectioning and Morphologic Analysis

Tissue sections from formaldehyde fixed paraffin embedded tumors were stained with hematoxylin and eosin (H&E). Histology of tumors were evaluated by a soft tissue pathologist (H.F). At least four tumors were analyzed for each treatment condition (Appendix A). Representative tumor sections were selected for figures.

### 2.6. Western Blot

Whole-cell extracts from tumor cell lines or tissue lysates from tumors were used. Whole-cell extracts from tumor cell lines were prepared from 6-well plates or T25/T75 flasks by scraping of cells in RIPA lysis buffer supplemented with 5 mM EDTA and 1× Halt protease and phosphatase inhibitor cocktail. Protein concentration measurement of samples were done using DC protein assay (Bio-Rad, Hercules, CA, USA). SDS-PAGE was performed using the NuPAGE system (Thermo Fisher Scientific, Waltham, MA, USA). Separated proteins were transferred to a PVDF membrane (Thermo Fisher Scientific, Waltham, MA, USA) and blocked with either 5% skim milk or 5% BSA in TBS-T buffer (50 mM Tris-HCL pH 6.8, 50 mM NaCl, 0.1% Tween 20). Membranes were incubated over night with primary antibody (Appendix A). After washes in TBS-T, secondary HRP-conjugated antibodies were added for 1 h incubation at room temperature. Chemiluminescence was detected using ImageQuant Amersham 800 upon incubation with SuperSignal West Dura Extended Duration substrate or SuperSignal West Femto Max Sensitivity Substrate (Thermo Fisher Scientific, Waltham, MA, USA). If detection of multiple bands with similar sizes on the same membrane was required, membranes were stripped using ReBlot Plus stripping buffer (Merck, Kenilworth, NJ, USA) and verified for successful stripping, before re-incubation with a new primary antibody. Bands were quantified using ImageJ.

### 2.7. Flow Cytometry

To analyze cell cycle distribution upon drug treatment, flow cytometry was performed after propidium iodide (PI) staining. Tumor cell lines treated with HSP90 inhibitors for 24 or 48 h were trypsinized and incubated in Vindelov’s solution (10 nM Tris pH 8.0, 10 mM NaCl, 0.1% NP-40, 700 U/L RNase, 75 µM PI) for 1 h at room temperature. Cell aggregates were removed by filtering through a 70 µm cell strainer (BD Bioscience, Franklin Lakes, NJ, USA). Flow cytometry was performed using a BD Accuri C6 flow cytometer (BD Biosciences, Franklin Lakes, NJ, USA) and data was analyzed using CFlow Plus software (BD Biosciences, Franklin Lakes, NJ, USA).

### 2.8. Phospho-RTK Array

Expression analysis of 49 RTKs was performed using Proteome Profiler Human Phospho-RTK Array Kit (R&D Systems, Minneapolis, MN, USA), according to the manufacturer’s protocol. For each sample, 300 µg of cell lysate was used. Relative expression of pRTKs in MLS cell lines 402-91 or 2645-94 treated with 100 nM 17-DMAG, 50 nM AUY922 or 80 nM STA-9090 for 24 h was calculated compared to untreated controls. ImageQuantTL (GE Healthcare, Chicago, IL, USA) was used for quantification and samples were normalized to average expression of all pRTKs in each array. Graphs were created using GraphPad Prism 9.1.0 (GraphPad, San Diego, CA, USA).

### 2.9. Reverse Transcription, PCR and Fragment Analyzer

Complementary DNA synthesis of RNA from MLS PDX tumor tissue was performed using the GrandScript cDNA synthesis kit (TATAA Biocenter, Gothenburg, Sweden) in 10 µL reactions containing 100 ng of RNA, 1× TATAA GrandScript RT reaction Mix and 1× TATAA GrandScript RT Enzyme (both TATAA Biocenter, Gothenburg, Sweden). Reverse transcription was performed in a T100 thermal cycler (Bio-Rad, Hercules, CA, USA) with a temperature profile of 22 °C for 5 min, 42 °C for 30 min, 85 °C for 5 min, followed by cooling at 4 °C until subsequent analysis. PCR was performed with primers targeting FUS-DDIT3 breakpoint 1 or breakpoint 2 (Appendix A). MLS tumor cell lines 402-91 (containing a type 1 fusion) and 2645-94 (containing a type 2 fusion) were used as positive controls. PCR was run in a CFX384 Touch Real-Time PCR detection system (Bio-Rad, Hercules, CA, USA) using 1× SYBR GrandMaster mix (TATAA Biocenter, Gothenburg, Sweden), 400 nM forward and reverse primers and 2 µL diluted cDNA in a final reaction volume of 6 µL. The PCR was run at 95 °C for 2 min, followed by 35 cycles amplification at 95 °C for 3 s, 58 °C for 30 s and 72 °C for 10 s, followed by melt-curve analysis. The PCR products were analyzed using Fragment Analyzer with the DNF-915 dsDNA reagent kit (Agilent Technologies, Santa Clara, CA, USA), according to the manufacturer’s instruction. PROSize3 software was used for data analysis.

## 3. Results

### 3.1. HSP90 Inhibition Affects Cell Viability of MLS Cell Lines In Vitro

Cell viability after HSP90 inhibition with 17-DMAG, AUY922 and STA-9090 was evaluated in vitro in MLS cell lines 402-91, 2645-94 and 1765-92. All three HSP90 inhibitors affected cellular viability, with equal or higher sensitivity than doxorubicin. AUY922 was effective at the lowest concentrations (Figure 1A–C,E). Normal human skin fibroblasts (F470), serving as non-neoplastic controls, were not affected by HSP90 inhibition whereas doxorubicin treatment led to a dose-dependent reduction in cellular viability (Figure 1D,E). In an attempt to determine the role of the fusion oncogene FUS-DDIT3 in the sensitivity to HSP90 inhibition, cell viability assays were also performed on the fibrosarcoma cell line HT1080 and the stably transfected HT1080-FUS-DDIT3-EGFP cell line. HT1080-FUS-DDIT3-EGFP and wildtype HT1080 had similar sensitivity to HSP90 inhibition (Appendix A). In summary, MLS cell lines are vulnerable to all three HSP90 inhibitors, whereas normal fibroblasts are not affected. No altered sensitivity to HSP90 inhibition was detected in HT1080 cells due to the presence of the fusion protein FUS-DDIT3.

### 3.2. HSP90 Inhibition Leads to a G2-M Arrest and Apoptosis

Cell cycle analysis using flow cytometry was performed on MLS tumor cell lines to further analyze the mechanism behind cell death mediated by HSP90 inhibitors. MLS 402-91, 2645-94 and 1765-92 were treated for 24 h with HSP90 inhibitors at doses slightly higher than IC50 values to ensure cytotoxic effects. Treatment with all HSP90 inhibitors led to a G2/M-arrest and G2/M accumulation (Figure 1F and Appendix A). Treatment with 17-DMAG for 24 and 48 h indicated that longer treatment time increased the level of G2/M-accumulation (Figure 1G and Appendix A).

Western blot analysis of protein extracts from the treated MLS cell lines showed that the three inhibitors had no effect on HSP90 levels but led to compensatory induction of the co-chaperone HSP70 (Figure 1H), commonly used as a pseudomarker for HSP90 inhibition [33]. This supports that HSP90 was targeted by all three HSP90 inhibitors. Increased cleavage of caspase 3, indicating activation of apoptosis, was observed upon treatment with all three HSP90 inhibitors (Figure 1H). The levels of fusion oncoprotein FUS-DDIT3 were not affected by the drugs, with the exception of 2645-94 where 17-DMAG treatment markedly reduced the level of FUS-DDIT3 (Figure 1H).

### 3.3. HSP90 Inhibition Affects Phosphorylation Levels of Receptor Tyrosine Kinases

HSP90 inhibition has been shown to affect the signaling of multiple RTKs in MLS and other malignancies [22,25,34]. To evaluate the effect of different HSP90 inhibitors on pRTK activity, pRTK arrays were employed. Since MLS cell lines 402-91 and 2645-94 express the two most common FUS-DDIT3 variants [35,36], these cell lines were used for analysis. HSP90 inhibition led to divergent patterns of pRTK activity in 402-91 and 2645-94 cells (Figure 2A,B, Appendix A). In 402-91, HSP90 inhibition with all three drugs led to a global increase in pRTK activity (Figure 2A). In 2645-94, HSP90 inhibition with all three drugs resulted in a global reduction of pRTK activity (Figure 2B). Twelve and twenty-seven pRTKs were downregulated by all three HSP90 inhibitors in 402-91 and 2645-94, respectively. Four of these—EGFR, ERBB3, INSR (insulin receptor) and EPHB3—were downregulated by all three HSP90 inhibitors in both 402-91 and 2645-94 (Figure 2C,D). EGFR and ERBB3 belong to the epidermal growth factor receptor tyrosine kinase family and signal mainly through the MAPK and PI3K/AKT signaling pathways [37] while INSR signals mainly through the PI3K/AKT pathway [38]. No pRTK was upregulated by all three HSP90 inhibitors in both cell lines.

### 3.4. Effects of HSP90 Inhibition on MAPK and PI3K/AKT Signaling Pathways

Western blot analysis was performed to verify the results obtained from the pRTK arrays and to further evaluate the downstream signaling of MAPK and PI3K/AKT pathways upon HSP90 inhibition. Treatment with all three HSP90 inhibitors led to a reduction of ERBB3 and EGFR RTK activity (Figure 3A), either by reduction of phosphorylation levels (ERBB3) or total protein expression levels (EGFR).

HSP90 inhibition generally led to a reduction of MAPK signaling, indicated by a reduction of phosphorylation and total protein expression levels of MEK1/2 and ERK (Figure 3A). The reduction was less prominent for AUY922, and in 402-91 and 1765-92, AUY922 led to increased phosphorylation of ERK (Figure 3A). HSP90 inhibition led to a reduction of AKT on both phosphorylation and total protein expression levels, with AUY922 exerting the lowest effect (Figure 3A). In conclusion, HSP90 inhibition led to reduction of MAPK and PI3K/AKT signaling. However, treatment with AUY922 inhibited these pathways to a lower extent and led to compensatory upregulation of downstream ERK activity in MLS cell lines 402-91 and 1765-92 (Figure 3B).

### 3.5. Effects of HSP90 Inhibitors on Tumor Growth In Vivo

Since all three HSP90 inhibitors showed a similar response in vitro, we followed up with comparative in vivo treatments of an MLS PDX model. This PDX model recapitulates the growth pattern and morphology of the original MLS tumor and was shown to harbor a type 2 FUS-DDIT3 fusion oncogene, the most common fusion oncogene variant in MLS [35] (Appendix A).

### 3.6. The Effect and Tolerability of the Different HSP90 Inhibitors Varied Considerably In Vivo

STA-9090 was initially planned to be given daily in a dose of 50 mg/kg, according to previous reports [39,40]. However, a primary evaluation using this dose for two weeks, indicated drug-related toxicity with significant weight loss of the mice. Consequently, STA-9090 was given in a lower and tolerable dose of 25 mg/kg, twice per week. This regime led to a transient arrest in tumor growth. However, after five weeks, the tumor volumes were equal to the control group (Figure 4A). Treatment with 17-DMAG led to a significant reduction in tumor growth (Figure 4A). On the contrary, AUY922 led to a small, non-significant increase in tumor volume (Figure 4A). The treatments, after dose-adjustment of STA-9090, were all tolerable and no significant weight loss was observed (Figure 4B). After five weeks, the mice were euthanized and tumor tissues were collected for further analysis.

Since the treatment response of AUY922 was poor in comparison to 17-DMAG, we tested the therapeutic effect of switching HSP90 inhibitor, from AUY922 to 17-DMAG. Mice that had been treated with AUY922 for five weeks had a treatment intermission for eight weeks and were subsequently rechallenged with 17-DMAG for three weeks. The control group consisted of mice previously treated with vehicle control only, that received vehicle control for three weeks. Rechallenge with 17-DMAG led to a reduction of tumor volume, also in mice previously treated with AUY922, although not reaching statistical significance (Figure 4C).

### 3.7. Morphological Effects of HSP90 Inhibitors In Vivo

A morphological analysis on treated PDX tumors was performed (Appendix A). Untreated MLS tumors had classic MLS features with tendencies of myxoid pools, thin-branched blood capillaries and occasional lipoblasts. The cellular density was low in central parts of the tumors, but higher in the peripheral zones. Furthermore, 17-DMAG-treated tumors were hypocellular, with a higher number of lipoblasts (Figure 4D). AUY922 treated tumors showed a focally increased atypia with increased cellular density both centrally in the tumors and in the border regions (Figure 4D). STA-9090-treated tumors resembled untreated MLS tumors, but with an increased number of vacuolized lipoblasts (Figure 4D). The tumors that were pretreated with AUY922 and rechallenged with 17-DMAG showed a similar treatment response as the tumors only treated with 17-DMAG (Figure 4E). The central parts of the tumors contained necrotic cells. The border region contained more viable cells of different sizes and shape but also necrotic cells, indicating a treatment response to the drug. Interestingly, tumors from mice that received toxic doses of STA-9090 for two weeks in the primary evaluation, showed a distinctive morphological pattern with a remarkably high degree of lipoblastic differentiation (Figure 4F).

### 3.8. Combination Treatment of HSP90 Inhibitors with Doxorubicin or Trabectidin

To examine the potential drug synergy between HSP90 inhibitors and doxorubicin or trabectidin in vitro, combination drug assays were set up. Combination treatment with doxorubicin and 17-DMAG, AUY922 or STA-9090 showed no drug synergy, with ZIP scores close to zero (Figure 5A,C,E and Appendix A). In contrast, trabectidin in combination with all three HSP90 inhibitors showed a drug synergy, with ZIP scores ranging from 6.2 to 7.2 (Figure 5B,D,F and Appendix A). In another experiment, combination treatment with HSP90 inhibitors and higher doses of trabectidin showed low level of drug synergy. However, at the highest trabectidin dose (10 nM), >99% of the cells were dead, which restricted the possibility to assess drug synergy. Even in this experiment, at lower doses of trabectidin, drug synergy was observed (Appendix A). The combination of trabectidin and doxorubicin also had synergistic effect (ZIP score 5.2), indicating that these drugs interact in a non-competitive way to promote cell death (Appendix A).

## 4. Discussion

HSP90 inhibition has been proposed as a potential targeted therapy for MLS [25,36]. Most HSP90 inhibitors target the ATP-binding pocket of HSP90 and could be expected to show similar effects [27]. Indeed, our results confirmed that the three HSP90 inhibitors 17-DMAG, AUY922 and STA-9090 have similar effects on cell viability in vitro (Appendix A). However, these drugs differ in effect on pRTK activity and downstream signaling in vitro and show a significant difference in treatment effects in vivo (Appendix A). Whereas 17-DMAG led to a continuous growth arrest, STA-9090 led to an early but transient growth arrest and AUY922 showed no inhibition of tumor growth. Furthermore, the morphological evaluation after AUY922 treatment indicated a transition to a more cellular and focally pleomorphic phenotype. Morphological response is closely related to survival in soft-tissue sarcoma. Patients with soft-tissue sarcoma who receive a complete pathological response upon neoadjuvant oncological treatment have a significantly better prognosis [41]. In MLS, morphological transitions, such as lipoblastic or adipocytic maturation, have been observed as a result of neoadjuvant treatment [42,43]. Considering that AUY922 failed to reduce tumor growth and led to a seemingly more aggressive morphology suggests that this drug would not be suitable to pursue into clinical trials against MLS.

The divergent in vivo responses are most likely related to upregulation of compensatory pathways, or by off-target effects of the drugs. Reactivation of ERK and AKT has been shown to drive drug resistance to AUY922 in BRAF mutated colon cancer [44] and to STA-9090 in KRAS-mutant non-small cell lung cancer [45]. Indeed, we observed that AUY922 led to an increase of ERK activity and failed to diminish AKT activity in MLS cell lines 402-91 and 1765-92, indicating a possible resistance mechanism. However, resistance could also be attributed to upregulation of other RTKs that have not been further investigated here. As observed in the pRTK array, AUY922 led to an upregulation of many pRTKs, particularly pRTKs belonging to the ephrin family of proteins (Appendix A).

The divergent in vivo results could also to a certain degree be attributed to suboptimal doses of the drugs. STA-9090 was possibly administered at a lower than optimal dose due to drug related toxicity. On the other hand, AUY922 was given in the same dose as previously reported by Steinmann et al. [25]. In parallel to our results, Steinmann et al. reported in vitro activity of AUY922 on MLS cell lines. In contrast to our results, AUY922 was reported to have effect in the treatment of an in vivo model [25]. The disparate results are probably explained by differences in the in vivo model. Steinmann et al. injected NOD/SCID mice with the SW872 tumor cell line derived from an unrelated sarcoma entity, “undifferentiated liposarcoma” with completely different genetic and biological features compared to MLS [46]. We instead used an MLS PDX model in nude mice, closely resembling morphological and clinical features of human MLS. Furthermore, the different genetic background of mouse strains might also have had an impact on the different outcomes.

Among the three tested HSP90 inhibitors, 17-DMAG showed the most promising clinical potential in vivo. STA-9090 showed promising morphological effects but was limited by dose-dependent drug toxicity in mice. AUY922 resulted in larger and morphologically more aggressive tumors. These results call for caution and a need for more thorough preclinical investigations before proceeding into clinical trials with AUY922 or other HSP90 inhibitors in patients with MLS.

Synergistic drug effects were observed in vitro between HSP90 inhibitors and trabectidin but not with doxorubicin. Trabectidin is a minor groove DNA double helix inhibitor but has also been suggested to have secondary mechanisms of action. For instance, trabectidin has been shown to inhibit the interaction between DNA and the FET-fusion oncogenes FUS-DDIT3 in myxoid liposarcoma and EWSR1-FLI1 in Ewing sarcoma [47]. Speculatively, the synergism might be due to a dual inhibition of the tumor-driving fusion oncoproteins and parallel inhibition of vital receptor tyrosine kinase signals necessary for MLS survival. However, the mechanism for drug synergy between HSP90 inhibitors and trabectidin needs to be further evaluated.

## 5. Conclusions

In conclusion, our results indicate that the in vivo effects of treatment with different HSP90 inhibitors differ significantly, which should be taken into consideration when further clinical trials are initiated with these compounds. The HSP90 inhibitor 17-DMAG showed the greatest in vivo effect in our MLS PDX model, whereas AUY922 showed a trend towards increased tumor growth and a more aggressive tumor morphology. In addition, combination treatment with HSP90 inhibitors and trabectidin showed synergistic effects in vitro and should be further investigated.

## Figures and Tables

**Figure 1 biomedicines-10-00624-f001:**
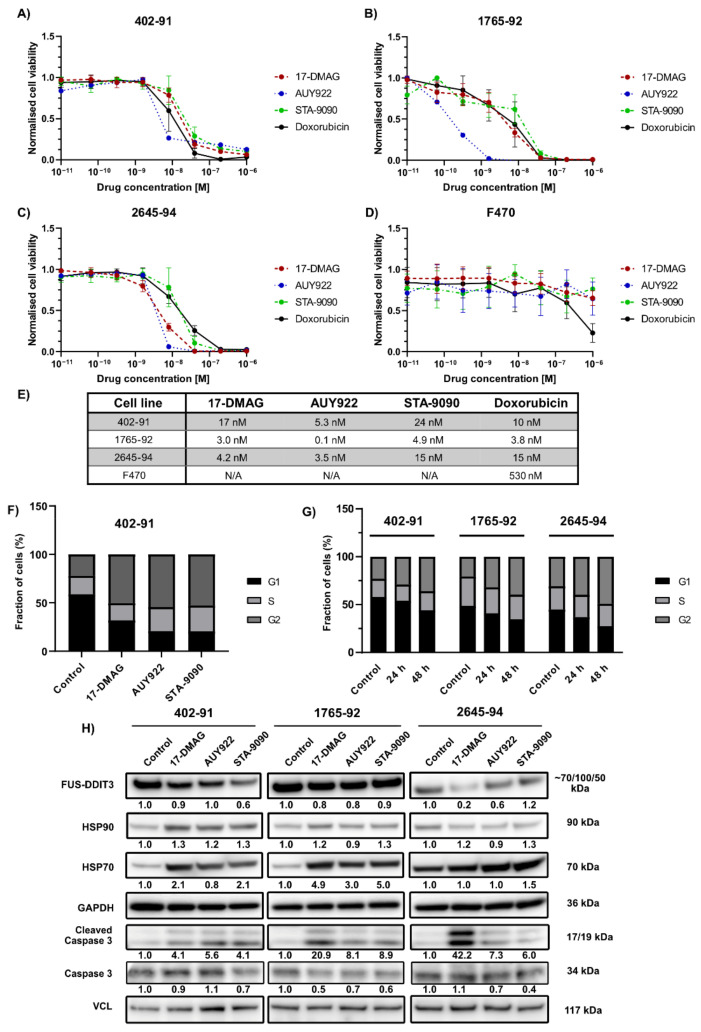
Effect of HSP90 inhibitors in vitro. (**A**–**D**) Cell viability assays of the HSP90 inhibitors 17-DMAG, AUY922 and STA-9090 and the chemotherapeutic agent doxorubicin on MLS cell lines 402-91 (**A**), 2645-94 (**B**) and 1765-92 (**C**) and human fibroblasts (F470) (**D**). Data is normalized to untreated controls. Mean ± SD is shown. (**E**) Table of IC50 values of tested compounds on MLS cell lines and human fibroblasts. N/A signifies IC50 value above maximum drug concentration (>1 µM). (**F**) Cell cycle analysis using flow cytometry on MLS cell line 402-91 treated with 100 nM 17-DMAG, 50 nM AUY922 or 80 nM STA-9090 for 24 h. (**G**) Cell cycle analysis using flow cytometry on MLS cell lines 402-91, 2645-94 and 1765-92, treated with 30 nM 17-DMAG for 24 or 48 h. (**H**) Western blot of whole-cell extracts from MLS cell lines 402-91, 1765-92 and 2645-94 treated with 100 nM 17-DMAG, 50 nM AUY922 or 80 nM STA-9090 for 24 h with antibodies against DDIT3 (targeting FUS-DDIT3), HSP90, HSP70, caspase 3, cleaved caspase 3 and VCL (vinculin). GAPDH was used as an internal protein loading control. The band intensities of target proteins were quantified by densitometric analysis and normalized to untreated controls. Normalized expression is shown in numbers below each blot. For double bands, both bands were included in the densitometric analysis.

**Figure 2 biomedicines-10-00624-f002:**
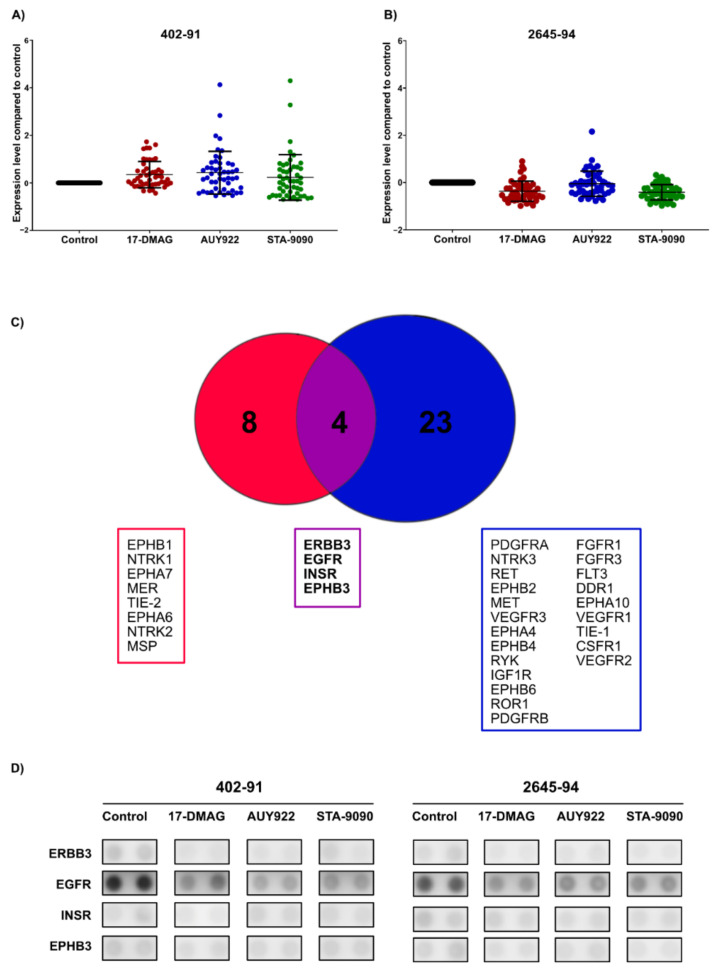
Effect on p-RTK activity upon HSP90 inhibition. Scatter plots of global pRTK expression upon HSP90 inhibition with 100 nM 17-DMAG, 50 nM AUY922 or 80 nM STA-9090 on MLS cell lines 402-91 (**A**) and 2645-94 (**B**). Each dot represents the normalized expression of a single pRTK. Mean expression ± SD is indicated for each treatment. (**C**) Venn diagram indicating pRTKs downregulated by all three HSP90 inhibitors in MLS 402-91 (red), MLS 2645-94 (blue) or in both cell lines (purple). No pRTKs were commonly upregulated by all three HSP90 inhibitors in both MLS 402-91 and 2645-94. (**D**) Visualization of the pRTKs from the pRTK membranes that were commonly downregulated by all three HSP90 inhibitors in both MLS 402-91 and MLS 2645-94.

**Figure 3 biomedicines-10-00624-f003:**
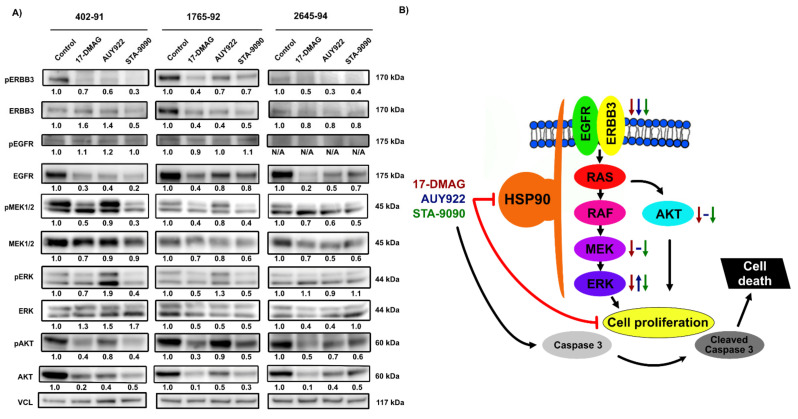
Downstream analysis on MAPK and PI3K/AKT signaling pathways. (**A**) Western blot analysis of ERBB3, EGFR and downstream signaling through MAPK and PI3K/AKT pathways on MLS cell lines 402-91, 1765-92 and 2645-94 treated with 100 nM 17-DMAG, 50 nM AUY922 or 80 nM STA-9090 for 24 h. VCL (vinculin) was used as an internal protein loading control. The band intensities of target proteins were quantified by densitometric analysis and normalized to untreated controls. Normalized expression is shown in numbers below each blot. For double bands, both bands were included in the densitometric analysis. (**B**) Illustration of affected signaling pathways after HSP90 inhibition with the effect of respective HSP90 inhibitor indicated by arrows.

**Figure 4 biomedicines-10-00624-f004:**
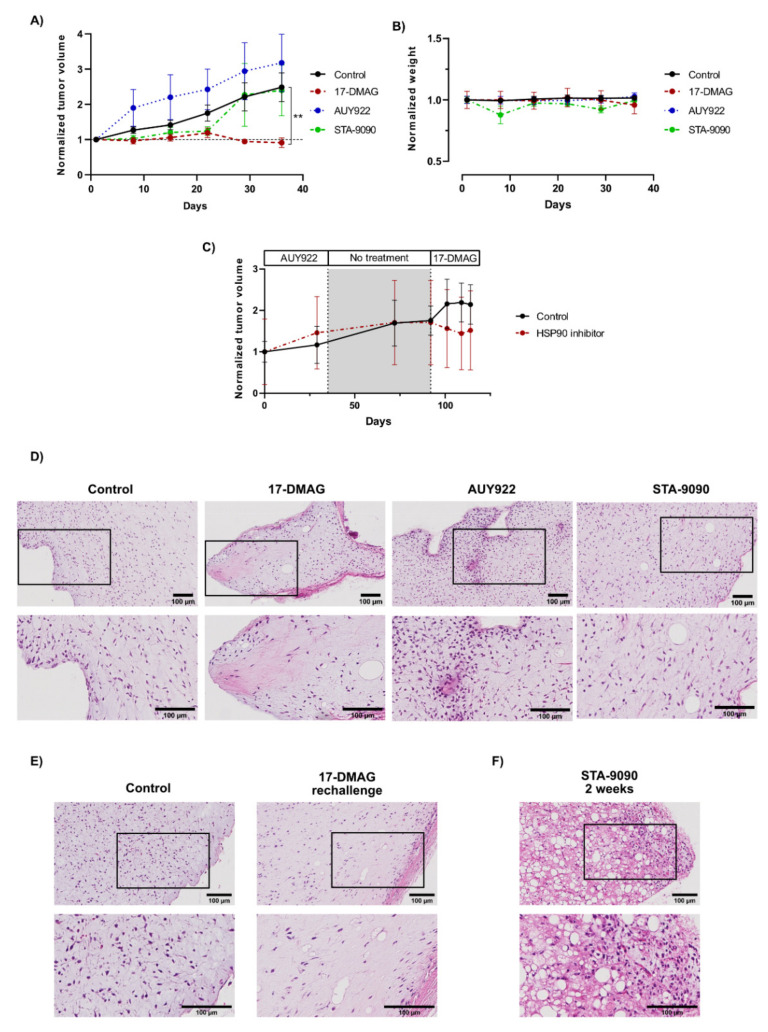
In vivo treatment of HSP90 inhibitors on an MLS PDX model. (**A**) Tumor growth normalized to baseline during a 5-week treatment course with HSP90 inhibitor 17-DMAG, AUY922, STA-9090 or vehicle control. All drugs were administered by intraperitoneal injection. Each treatment group consisted of at least 8 tumors. Data is presented as mean ± SEM. Statistical analysis was performed using Kruskal–Willis test with multiple comparison correction using Dunn’s test. Significance was calculated using *p*-value, where *p* < 0.05 was considered significant (** = *p* < 0.01, no marker = n.s). (**B**) Normalized weight of mice during treatment. Data is shown as mean ± SEM. (**C**) Tumor growth normalized to baseline upon rechallenge with 17-DMAG in mice pretreated with AUY922. Control group consisted of mice previously receiving vehicle control, treated with vehicle control upon rechallenge of drug treatment. Initial treatment was given for 5 weeks, followed by 8 weeks of treatment intermission, followed by 3 weeks of rechallenge treatment. Each treatment group consisted of at least 4 tumors. Data is shown as mean ± SEM. (**D**–**F**) H&E staining of representative tumors, treated with indicated drugs. The lower image is an enlargement of the area indicated by the frame in the upper image.

**Figure 5 biomedicines-10-00624-f005:**
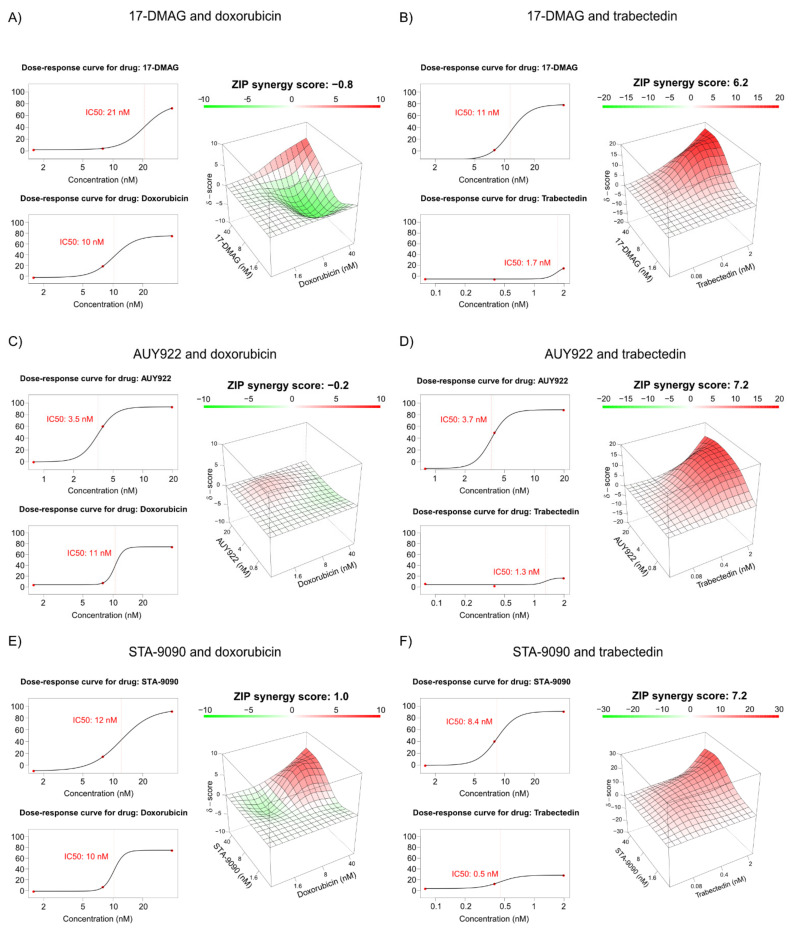
Combination cell viability assays. (**A**–**F**) Dose-response curves and synergy matrixes with mean ZIP synergy score of MLS 402-91 treated with HSP90 inhibitors and doxorubicin (**A**,**C**,**E**) or trabectedin (**B**,**D**,**F**) at indicated doses. Dose-response curves display inhibition of cell viability upon indicated concentration of each drug. IC50 values for each individual drug are indicated in red. The synergy matrixes plot ZIP synergy scores for each dose combination of the drugs. A mean ZIP synergy score is displayed above each matrix. Negative values of ZIP score indicate antagonism, 0 indicates additive drug effect and positive values indicate synergistic effect.

## Data Availability

Not applicable.

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
