# Peer review of "Different HSP90 Inhibitors Exert Divergent Effect on Myxoid Liposarcoma In Vitro and In Vivo"

_biomedicines, 2022, doi:10.3390/biomedicines10030624_

Round 1

Reviewer 1 Report

In this manuscript, Vannas and coworkers performed in vitro and vivo studies aimed at testing 3 different HSP90 inhibitors, namely 17-DMAG, AUY922 and STA-9090, as possible therapeutic options for patients with Myxoid Liposarcoma (MLS). To perform in vitro studies, authors used 402-91, 1765-923 and 2645-94 MLS cell lines. Results showed that inhibitors induce a G2/M arrest of the cells and increase levels of cleaved-caspase 3. In these experiments, AUY922 resulted the inhibitor more effective at lowest concentration. Then authors tested the effect of the 3 inhibitors on pRTK levels on the 2 MLS cell lines carrying the most common FUS-DDIT3 variants. Results showed that levels of 4 pRTK, EGFR, ERBB3, INSR, EPHB3 are commonly down-regulated in both cell lines. In addition, authors showed that HSP90 inhibitors induce a reduction of MAPK and PI3K/AKT signaling pathways, with AUY922 showing the lowest efficacy in the down-regulation of these pathways. In the last part of the manuscript, authors performed in vivo experiment with a PDX model of MLS. Results showed that treatment with 17-DMAG reduces tumor growth, while treatment with AUY922 inhibitor induce an increase in tumor growth that could be reverted by the switch to treatment with 17-DMG. In vitro combination drug assays revealed a drug synergy between HSP90 inhibitors and trabectidin.

Data here presented are of interest, but, in my opinion, the manuscript is too descriptive and several major issues should be addressed:

-Main results of the in vitro experiments should be summarized in a table, along with molecular characterization of the MLS cell lines, and discussed for their potential clinical applications. Why did the authors tested for pRTK levels only the 2 MLS cell lines with the most common FUS-DDIT3 variants, but validate results in all 3 MLS cell lines?

-Authors should explain and discuss why they reported how HSP90 inhibitors affect MLS morphology. Are those tumor morphology features related to tumor aggressiveness or patient clinical outcome?

Minor issues:

-Please add statistical analysis where replicates have been tested.

-Figure 3B is not cited in the text, please add it.

-Why did the authors tested 2 doses of trabectidin in combination drug assays?

-Please verify the number of the Supplementary Figures cited in the last paragraph ‘Combination treatment of HSP90 inhibitors with Doxorubicin or Trabectidin’.

Reviewer 2 Report

line 314: route of administration is not reported

from line 348 through 354 before the full-stop: move to introduction

line 349-50 substitute "potency" with "activity"

une 359: omit "interestingly" since the synergy is known 

line 401 substitute "efficacy" with "activity"

line 427: substitute "instigated" with "initiated"

Round 2

Reviewer 1 Report

The authors have adequately addressed all my concerns and queries. I have only a minor comment: Authors should better specify the differences between the FUS-DDIT3 variants (type 1, type 2 and type 6 fusions) of the MLS cell lines of the study.